# COVID-19 Vaccination Personas in Syria: Evidence from a Cross-Sectional Survey

**DOI:** 10.3390/vaccines11061109

**Published:** 2023-06-16

**Authors:** Zlatko Nikoloski, Elnur Aliyev, Robert E. S. Bain, Leonardo Menchini, Sahar Hegazi, Mai Zalkha, Shaza Mouawad, Neha Kapil, Amaya M. Gillespie

**Affiliations:** 1Department of Health Policy, London School of Economics and Political Science, London WC2A 2AE, UK; 2UNICEF Syria Office, Damascus P.O. Box 9413, Syria; 3Regional Office for the Middle East and North Africa, Amman 11821, Jordan

**Keywords:** COVID-19, demand, personas, vaccine hesitancy, vaccine acceptance

## Abstract

Achieving a high level of COVID-19 vaccination coverage in a conflict-affected setting is challenging. The objective of this paper is to shed further light on the main determinants of vaccination coverage using a large, cross-sectional sample (October–November 2022) of over 17,000 adults in Syria. We find evidence that certain demographic and socioeconomic characteristics describe a core set of vaccination personas. Men, older respondents, and those who are more educated and trust information received from healthcare authorities are more likely to be vaccinated. Healthcare workers in this sample are highly vaccinated. Furthermore, respondents with more positive views towards COVID-19 vaccines are also more likely to be willing to be vaccinated. By contrast, respondents who believe that vaccines are associated with significant side effects are also more likely to refuse vaccination. In addition, younger respondents and women, as well as those with a lower level of education, are more likely to refuse to be vaccinated. Respondents with a neutral attitude towards vaccines are also more likely to be undecided, whereas respondents who are refusing to get vaccinated are more likely to trust the information received from private doctors, private clinics, as well as social media and, more broadly, the internet.

## 1. Introduction

For more than 12 years, Syria has been affected by conflict, with 14.6 million people in need of humanitarian assistance, including 6.9 million children, 4.2 million people with disabilities, and 5.3 million internally displaced persons (IDPs) by the end of 2022 [1]. Since the beginning of the conflict, at least 11 million people have fled their homes, most of them remaining in neighboring countries. In February 2023, a devastating earthquake came on top of a worsening economic crisis, disease outbreaks, localized hostilities, mass displacement, and weak public infrastructure.

Against this backdrop, Syria has also weathered the effects of COVID-19. The vaccination drive in Syria commenced by late 2021, when over a quarter of a million doses of the Astra Zeneca vaccines were welcomed into the country [2]. As in the rest of the world, the initial drive gave preference to the most at-risk groups: healthcare workers, people suffering from NCDs (noncommunicable diseases), as well as the elderly. Despite the success associated with the initial vaccination drives, pockets of vaccine hesitancy were evident across the country [2].

To date, only a few studies have tried to distil some of the main correlates of COVID-19 vaccination status in Syria, while also shedding some light on the potential barriers that prevent the uptake of vaccines. In general, these studies are small, one-off, cross-sectional survey studies that focus on the main correlates of vaccination intention. A study on willingness to receive a COVID-19 vaccination [3] revealed that gender, age, residence, as well as significant knowledge associated with COVID-19 are the main socio-economic and demographic correlates of willingness to be vaccinated. Similar findings emerged from a second study, which also revealed that attitudes and beliefs regarding the COVID-19 vaccines (e.g., fear of side effects and belief in the effectiveness of the vaccines) were also a contributing factor towards one’s willingness to be vaccinated [4]. These findings have been corroborated by a regional study, where Syrians were a fraction of the sample (which also included Palestinians and Jordanians) [5], as well as in a study which used qualitative research methods [6].

Most recently, additional evidence has emerged regarding specific groups of the population or regarding additional shots (boosters) of the COVID-19 vaccine [7,8]. Interestingly, one of those studies found that Syrian refugees in Jordan showed a high vaccine acceptance rate, while also highlighting the importance of knowledge and awareness of the COVID-19 vaccine, the virus, and the disease to increase the acceptance rate [7]. Most recently, a study by Abouzid et al. [8], which encapsulated the findings from above, indicated that socioeconomic and demographic characteristics (e.g., age, gender) as well as beliefs and attitudes towards the vaccines (e.g., fear of side effects and beliefs around the effectiveness) are the main correlates of vaccine acceptance and vaccine hesitancy [8].

Against this background, the objective of this paper is to shed further light on the main determinants of vaccination coverage, using a large, cross-sectional sample of over 17,000 adults in Syria.

## 2. Methodology

### 2.1. Survey Instrument

We administered a cross-sectional survey on attitudes and beliefs towards COVID-19 vaccines in Syria between October and November 2022 on a sample of 17,000 adults. Subjects eligible for receiving the COVID-19 vaccine were randomly selected and enrolled in the study. Participants were mainly front-line workers from the health (health workers) and education sectors (teachers), randomly selected from healthcare centers and schools. In addition, key informants were selected from employees from related ministries. With that said, the survey relied on convenience sampling, meaning participants available at the time of the visit and eligible to receive the vaccine. The structured questionnaire used for data collection contained mainly multiple-choice, closed-ended questions, with limited open-ended questions. Data were collected face to face upon visiting healthcare facilities, schools, or government departments. A further breakdown of the sample size is provided in the Results section.

### 2.2. Data Analysis

This study is purely descriptive in nature, whereby we try to describe the main characteristics of the vaccination personas. In order to do that, our main outcome variable is a categorical variable, which captures the four vaccination personas: (a) those vaccinated; (b) those willing to be vaccinated; (c) those undecided; and (d) those not willing to be vaccinated. In order to distil the four personas, we relied on the following questions from the survey: “Have you received one or more COVID-19 vaccines?” and “How much would you like to receive the COVID-19 vaccine?”. This outcome variable was then used in a cross-tabulation exercise, whereby the four vaccination personas were described along three sets of independent variables: (i) demographic and socioeconomic characteristics (e.g., age, gender, and occupation,); (ii) a second group relating to beliefs towards the vaccines (e.g., beliefs in the vaccine safety and the vaccine’s side effects); (iii) the final group of characteristics corresponding to the communication channels used to reach the communities (e.g., receiving information about COVID-19 vaccines from the most trusted source of information).

The analysis used in this paper was descriptive in nature. The outcome variable was tabulated across the three sets of independent variables mentioned above in order to provide an understanding of the link between various individual characteristics and willingness to vaccinate. The cross-tabulated percentages were compared using Fisher’s exact test and the *p*-values were reported.

Finally, as a robustness check, we also conducted a preliminary logit modelling analysis. Four binary independent variables corresponding to the four vaccination personas were used in four separate logit modelling analyses. The four dependent variables were regressed against a battery of independent variables including socio-demographic variables as well as attitudes and beliefs regarding COVID-19 vaccines.

All analyses were conducted using Stata 17.

## 3. Results

We commence the results section with a brief overview of the descriptive statistics, which are captured in Table 1. First, the sample was significantly biased towards women. More specifically, about two thirds of the sample (65.7%) were women, while a third were men—reflecting the strong representation of women in health and education. In terms of age, a vast majority of the respondents (over 90%) were between the ages of 18 and 55. Unfortunately, given the survey instrument employed in the data collection, we were unable to disaggregate the sample further (e.g., in ten-year intervals). A total of 79% of the respondents in the sample were working; of those who were employed, the sample was heavily biased towards health workers. More specifically, 40% of the employed sample respondents were healthcare workers, while another 22% were teachers. Overall, the sample was well educated. About one third of the sample respondents had a secondary education, while another third of the respondents had a university degree or above. Only 2.5 percent of the sample respondents were illiterate. Furthermore, close to half of the respondents (44.9%) trusted all of the vaccines, while another 40% trusted only a few vaccines. Finally, the descriptive statistics table captures the vaccination status of the respondents. A large majority of the sample (77.5%) were vaccinated; 3.3% had not been vaccinated but were willing to do so, while 6.5% were not vaccinated and undecided. Finally, vaccine refusal was reported to be at 12.7%. In other words, about one in ten respondents from the sample were not vaccinated and not willing to be vaccinated.

In the following few sections, we provide a detailed account of the main characteristics of each of the four vaccination personas.

### 3.1. Vaccinated

A higher share of men (79.5%) compared to women (76.5%) tended to be vaccinated (Table 2). As the elderly were targeted in the first waves of the vaccination campaigns, it is not surprising that close to four in five respondents over the age of 55 were vaccinated (the share among those aged 18–55 was 71.9%). Additionally, a significant majority of those working (81%) were vaccinated—most likely driven by the workplace vaccination requirements. While 94% of the healthcare workers were vaccinated, 78% of ministry employees and about two thirds of teachers had received a vaccination. Finally, there was logarithmic link between the level of education and being vaccinated. In other words, the likelihood of being vaccinated increased up to secondary education level, where it plateaued (80.5% of those with secondary education and 78% of those with tertiary education).

Receiving information about COVID-19 had a positive effect on vaccination status in this sample. The majority of respondents (90.6%) who reported receiving information all the time stated that they were vaccinated. Furthermore, as the results reveal, more positive beliefs about the vaccines were associated with a higher probability of being vaccinated. For example, 94.6% of respondents who strongly believed that the vaccines were safe had been vaccinated. The same relationship held for the rest of the beliefs included in this survey: feeling at risk when obtaining a COVID-19 vaccination, trusting the vaccines, believing in the fairness of the vaccines, importance of the vaccines for one’s health, as well as the importance of the vaccines for the health of the wider community (Table 3).

Higher trust in the healthcare community was associated with a higher probability of being vaccinated. Most respondents (80%) with high trust in healthcare workers were also vaccinated (Table 3). This high trust in healthcare workers may explain why most of those vaccinated tended to receive their COVID-19-related information from healthcare staff. As Table 4 reveals, three quarters of those who had been vaccinated stated that they received their COVID-19-related information from healthcare staff.

### 3.2. Not Vaccinated but Willing

There were small differences in terms of age and gender among those not vaccinated but willing (Table 1). However, twice as many respondents who were not working (5.7%) compared to those who were working were willing to receive a COVID-19 vaccination. A slightly higher share of those with preparatory and primary education (compared to higher levels of education) were willing to be vaccinated (in part because most of those with higher education had already received at least one vaccine). While no discernible link emerged between those willing to be vaccinated and vaccination beliefs, the results from the analysis (Table 2) reveal that respondents who believed in the importance of the vaccines and in the protection they provide to family and community were more likely to be willing to be vaccinated. Similar to those who already had been vaccinated, those who were willing tended to receive their COVID-19 information from healthcare staff (Table 4).

Finally, as Table 5 reveals, those who were willing to be vaccinated reported a need for more information about the vaccines related to side effects and safety. More specifically, 45.5% of this group of people stated they would like to receive more information on side effects, while another third stated they would like to improve their knowledge regarding the safety of the vaccines. Resuming normal travel along with an increase in the feeling of being protected were considered as the main benefits of COVID-19 vaccination among this vaccination persona.

### 3.3. Not Vaccinated and Undecided

Demographically, there was no strong link between correlates such as age and gender with this vaccination persona. A slightly higher share of teachers (9%) compared to the health workers and ministry employees in the sample tended to be unvaccinated and undecided. Similarly, about one in ten respondents with primary education were likely to be undecided regarding receiving a COVID-19 vaccination.

The most important characteristic of this type of vaccination persona was their neutral beliefs regarding COVID-19 vaccines (Table 3). For example, one in five respondents who stated “3” (on a scale of 1 to 5, with 1 being the lowest and 5 being the highest) on the question about safety of the vaccines was also not vaccinated and undecided. Similar findings emerged regarding some of the other beliefs surveyed: being at risk when receiving a vaccination, the importance of the vaccines for one’s health, as well as the belief in the importance of the vaccine for the health of the wider community.

Unlike the other two personas mentioned above, those who were undecided about being vaccinated trusted healthcare workers less and relied more on social media (40%) when receiving COVID-19-related information (Table 4). They also tended to trust private doctors and clinics. In particular, 44.6% stated they would like to know more about side effects, while 34.3% stated they would like to receive more information about vaccine safety (Table 5).

### 3.4. Not Vaccinated and Unwilling

There were some clear empirical regularities between demographic characteristics and this vaccination persona. On average, a higher share of women (13.5%) compared to men (11%) were unwilling to be vaccinated. Those who were unwilling tended to be younger (Table 2). In addition, a higher share of those with a lower level of education were unwilling to be vaccinated. A total of 15.9% of those who were illiterate and 17.4% of those with only primary education were unwilling to receive a COVID-19 vaccination.

This vaccination persona was less informed about the vaccines and they did not trust healthcare workers. For example, a quarter of respondents who had never received information about COVID-19 were unwilling to be vaccinated (Table 3). In addition, this group of respondents held strong negative views regarding the COVID-19 vaccines. For example, 51.8% of those who believed that the vaccines were not safe tended to refuse them. Similar patterns emerged regarding the rest of the vaccination beliefs: feeling at risk when obtaining a COVID-19 vaccination, trusting the vaccines, believing in the fairness of the vaccines, importance of the vaccines for one’s health, as well as the importance of the vaccines for the health of the wider community (Table 3).

Finally, Table 5 provides a snapshot of reported information needs by persona. Mirroring the summaries on vaccination beliefs, it appears that receiving more information on vaccines’ safety and side effects could help persuade some of those who are undecided as well as the unwilling to receive the vaccination. For example, nearly half of those who were undecided said they would like to receive more information about the side effects of the vaccines (the share was equally high among those refusing the vaccines). Similarly high shares of respondents from the two groups mentioned that they needed additional information on the safety and effectiveness of the vaccines. In addition, as the lower part of the table attests, this vaccination persona was motivated by factors outside health benefits, such as the ability to travel, which reinforces the need for people-centered research to understand the issues from the community perspective.

As indicated in the Methodology section, we also conducted a robustness check, whereby four separate logit models (for the four different types of personas) were used to distill the main characteristics of the four different types of personas. The results (captured in the Appendix A
Table A1, Table A2, Table A3 and Table A4) unequivocally point out the established regularity from above—namely, that positive beliefs around COVID-19 vaccines are associated with a higher probability of being or willing to be vaccinated, while when negative beliefs around the vaccine increase, so does the probability of being vaccine-hesitant person.

## 4. Discussion

There was a three-fold objective that we pursued in this research paper: (i) first, to describe the demographic and socioeconomic characteristics of the four types of vaccination personas; (ii) to analyze the link between beliefs associated with COVID-19 vaccines and the four types of vaccination personas (vaccinated, willing to be vaccinated, undecided, and not willing); and (iii) analyze the potential platforms that could be used in order to increase vaccine acceptance and improve vaccine coverage in Syria. There are a few main findings that stem from this analysis. We found evidence that certain demographic and socioeconomic characteristics describe the vaccination personas. Men, respondents who are more educated, respondents who are older, as well as healthcare workers are more likely to be already vaccinated. Respondents who trust information received from the healthcare authorities are also more likely to be vaccinated. By contrast, those with a lower level of education, women, and younger respondents are more likely to refuse the vaccine. Respondents with more positive views towards COVID-19 vaccines are also more willing to be vaccinated. By contrast, those who are more likely to refuse vaccination tend to believe that vaccines are associated with significant side effects and tend to trust the information received from private doctors, private clinics, as well as social media and, more broadly, the internet. In addition, respondents with a neutral attitude towards vaccines are more likely to be undecided.

Our findings on the demographic characteristics of vaccination willingness are consistent with the existing evidence [3]. A recent paper using two waves of repeated cross-sectional surveys from the Middle East and North Africa (MENA) region [9], for example, found that men, on average, were more likely to be vaccinated and to be willing to be vaccinated, especially early in the pandemic when vaccine supplies were more limited. Moreover, a study by Abouzid et al. (2022) found that men were significantly more likely than women to receive a booster dose of the COVID-19 vaccine [8]. In the context of the MENA region, a range of individual, social, and structural factors have been identified as important influences on this pattern. In many settings, women face restrictive social norms that limit decision-making power and mobility and imply a heavy burden of care—all of which can affect the priority of vaccination as well as the practical aspects of accessing sites. In addition, men are also more likely than women to be in formal work in the MENA region and may have had more motivation from employers to be vaccinated [10].

One of the principal findings relates to the link between beliefs about vaccination and willingness to be vaccinated, which aligns with the considerable body of existing evidence from the MENA region. A study about vaccination among healthcare workers in Egypt, for example, found that the reasons for vaccine acceptance revolve around safety and effectiveness, while fear of side effects was the main reason for vaccine hesitancy. Concerns about safety as well as general lack of trust in the vaccines were the main reasons for vaccine hesitancy among healthcare workers in Sudan and Iraq [11,12]. Lack of trust in vaccine effectiveness and fear of side effects were the main reasons for refusing to be vaccinated also among the general population [9,13,14,15], while the belief in the effectiveness and benefits associated with the COVID-19 vaccination was the main reason for vaccine acceptance [14,16]. Beliefs associated with vaccination status are also the core determinants of vaccination status in Syria. Fear of possible side effects was the main reason for the reluctance to take the vaccine, followed by mistrust of the vaccine formula in a study conducted by Shibani et al. [4]. For women in the MENA region, there is evidence that specific types of misinformation, for example regarding fertility and vaccination, may have had an outsized effect on women’s acceptance of vaccines [10].

There are some limitations associated with this research. First, the analysis is descriptive and correlational in nature and it only establishes a correlation between vaccination status and the variables of interest. In that respect, we cannot infer any direct causal links by using this methodological approach. Second, the 17,000 respondents, while randomly selected from visitors to healthcare facilities, teachers at school and employees of government agencies could not be counted as providing a representative set of adults in the country. Furthermore, many of the factors studied here are highly fluid in that perspectives can change quickly. Thus, this study should be treated as descriptive and exploratory in nature according to the conditions at a certain point in time (October–November 2022).

These limitations notwithstanding, there are some broad conclusions and practical recommendations that stem from this research. First, given the differences observed, efforts should focus on tailoring responses to the different persona categories. Furthermore, those who are undecided or willing to be vaccinated (who tend to be a majority) are more likely to change their view in the short term, rather than those who are unwilling [17]. In addition, reframing the goal around increasing vaccination acceptance, rather than hesitancy, may also be useful in that it reinforces the positive outcome rather than the negative [17]. Our findings also show that motivation for vaccination goes beyond health-related benefits. For examples, access to travel was a significant motivator. This suggests that further exploration of motivations relevant to different personas will improve intervention results.

Beliefs around the COVID-19 vaccines were also linked with the willingness (or lack of willingness) to obtain the vaccination, and neutral attitude towards aspects of the vaccine (e.g., safety, effectiveness, and side effects) were associated with a higher likelihood of being undecided. The findings also suggest that interventions should take into account gender and education attainment when addressing these factors. For example, experience in Sudan, where interventions were designed to appeal to, improved uptake by making structural changes to vaccination sites, such as providing female vaccinators and additional privacy, as well as addressing misinformation and specific concerns, such as vaccination during pregnancy or for nursing mothers and effects on menstruation and fertility [10].

Providing consistent access to reliable trusted information channels can help in allaying fears and increasing confidence in the vaccines; however, the channels tend to vary for different persona groups. For example, those who rely on information from healthcare providers are more likely to be vaccinated, whereas vaccine hesitant individuals rely on information from private healthcare providers, social media, and, more broadly, the internet. Where information is carefully tailored to the needs of different personas, the effect is more likely to be positive, e.g., addressing specific misinformation about fertility is likely to influence women. A review of studies in MENA reiterates the centrality of religion in the region; however, the correlation with vaccine acceptance can be positive or negative [18]. In many countries, religious leaders strongly endorsed vaccination, including by offering places of workshop as vaccination sites. At the same time, religious concerns were raised about the contents of vaccines or whether vaccination during Ramadan should be allowed [18]. Taken together, these findings reinforce the need for various “influencers” to be well trained and supported to use different approaches when interacting with different personas.

The use of personas is a practical way to tailor and localize responses to the needs of communities, rather than applying a “one size fits all” approach. While this study focuses on COVID-19 vaccination, the implications for this type of behavioral analysis to form personas also applies to other behaviors or outbreaks, especially for protracted situations where “response fatigue” can be a challenge.

## Figures and Tables

**Table 1 vaccines-11-01109-t001:** Syria COVID-19 vaccination status survey: descriptive statistics.

	%	N
Gender		
Female	65.7	11,192
Male	34.3	5839
Age		
18–55	91.6	15,601
55 or more	8.4	1430
Working		
No	21.0	3570
Yes	79.0	13,461
Type of work		
Employee	26.4	3549
Health worker	40.0	5377
Other	11.8	1592
Teacher	21.9	2943
Education level		
Illiterate	2.5	421
Preparatory	20.1	3420
Primary	8.1	1387
Secondary	35.0	5966
University and above	34.3	5837
Trust in the vaccines		
I do not trust vaccines	14.3	2091
I only trust a few vaccines	40.8	5953
I trust all vaccines	44.9	6560
Vaccination status		
Vaccinated	77.5	12,864
Not vaccinated but willing	3.3	541
Not vaccinated and undecided	6.5	1082
Not vaccinated and unwilling	12.7	2103

**Table 2 vaccines-11-01109-t002:** Demographic characteristics and vaccination status.

	Vaccinated	Willing	Undecided	Not Willing	Fisher’s Exact Test *p*-Value
Male	79.5	2.9	6.6	11.0	<0.0001
Female	76.5	3.5	6.5	13.5	
	Vaccinated	Willing	Undecided	Not willing	
18–55	71.9	4.8	8.3	15.0	<0.0001
Over 55	78.1	3.1	6.4	12.5	
	Vaccinated	Willing	Undecided	Not willing	
Not working	64.6	5.3	10.8	19.4	<0.0001
Working	81.0	2.7	5.4	10.9	
	Vaccinated	Willing	Undecided	Not willing	
Health worker	94.0	1.3	2.1	2.7	<0.0001
Teacher	64.1	4.4	9.0	22.5	
Employee	78.0	3.5	6.3	12.2	
Other	73.2	3.0	8.4	15.3	
	Vaccinated	Willing	Undecided	Not willing	
Illiterate	74.1	4.2	5.9	15.9	<0.0001
Preparatory	76.2	3.9	7.2	12.7	
Primary	66.5	5.6	10.4	17.4	
Secondary	80.5	2.7	6.0	10.8	
University and above	78.2	2.8	5.8	13.3	

**Table 3 vaccines-11-01109-t003:** Beliefs and attitudes towards the COVID-19 vaccines and vaccination status.

Receiving Information about COVID-19					Fisher’s Exact Test *p*-Value
	Vaccinated	Willing	Undecided	Not willing	
Never	56.2	4.2	12.7	26.9	<0.0001
Sometimes	80.3	3.1	5.9	10.7	
Often	85.8	3.1	3.8	7.3	
All the time	90.6	2.2	2.5	4.7	
How challenging is to receive the COVID-19 vaccine?					
	Vaccinated	Willing	Undecided	Not willing	
1	77.6	3.1	6.1	13.2	<0.0001
2	73.0	3.4	7.9	15.8	
3	71.1	4.2	11.3	13.4	
4	84.9	3.1	4.7	7.3	
5	86.3	3.1	2.4	8.2	
How safe do you think the COVID-19 vaccines are?					
	Vaccinated	Willing	Undecided	Not willing	
1	41.9	2.7	3.6	51.8	<0.0001
2	45.6	1.9	10.9	41.6	
3	67.0	3.3	18.3	11.4	
4	90.9	4.6	2.8	1.8	
5	94.4	3.0	1.0	1.6	
Do you think you are at risk when taking the COVID-19 vaccines?					
	Vaccinated	Willing	Undecided	Not willing	
1	42.3	2.2	4.8	50.7	<0.0001
2	48.9	1.7	9.3	40.2	
3	69.0	3.6	17.5	10.0	
4	90.2	4.7	3.0	2.1	
5	94.6	2.9	1.0	1.5	
Trust in the COVID-19 vaccines					
	Vaccinated	Willing	Undecided	Not willing	
I do not trust vaccines	32.1	3.6	11.9	52.4	<0.0001
I only trust a few vaccines	85.3	3.2	6.6	4.9	
I trust all vaccines	95.1	2.0	1.3	1.6	
Trust in the healthcare workers					
	Vaccinated	Willing	Undecided	Not willing	
1	55.9	2.9	3.1	38.1	<0.0001
2	51.4	2.7	7.8	38.2	
3	53.3	3.6	17.3	25.8	
4	75.7	3.9	7.5	12.9	
5	79.6	4.1	6.1	10.2	
Fairness in distribution of the vaccines					
	Vaccinated	Willing	Undecided	Not willing	
1	65.3	1.9	4.0	28.9	<0.0001
2	59.1	2.3	8.5	30.1	
3	66.9	3.2	13.6	16.3	
4	79.8	3.9	6.1	10.2	
5	85.4	3.3	3.9	7.4	
Importance of the vaccines for one’s health					
	Vaccinated	Willing	Undecided	Not willing	
1	35.5	0.8	1.5	62.2	<0.0001
2	43.4	1.4	8.1	47.1	
3	63.6	2.9	20.0	13.6	
4	87.3	5.4	4.6	2.6	
5	93.1	3.1	1.9	1.9	
Protection offered to family and community					
	Vaccinated	Willing	Undecided	Not willing	
1	29.1	1.1	1.2	68.6	<0.0001
2	39.7	1.0	7.8	51.5	
3	63.7	2.2	24.7	9.4	
4	90.6	5.1	3.0	1.3	
5	95.3	3.6	0.5	0.6	

**Table 4 vaccines-11-01109-t004:** Vaccination status and sources of information.

Source of Info					Fisher’s Exact Test *p*-Value
	Vaccinated	Willing	Undecided	Not willing	
TV	28.4	36.0	36.1	33.6	<0.0001
Radio	3.8	5.6	4.9	3.4	0.037
Health staff	74.1	59.7	49.8	43.2	<0.0001
Community health workers	12.8	14.2	8.1	6.9	<0.0001
Peers	11.1	9.6	11.7	14.7	<0.0001
Community leaders	5.2	5.6	5.7	3.8	0.001
Social media	39.9	36.0	43.0	43.3	<0.0001
SMS	5.3	6.3	6.0	4.5	0.158
Family	7.3	10.0	10.5	14.9	<0.0001
At work	12.4	5.6	6.5	6.7	<0.0001
Internet	19.5	16.3	20.0	21.4	0.043
Private doctors/clinics	14.4	10.9	16.3	12.6	0.004
No answer	0.1	0.0	0.0	0.6	<0.0001
**Most trusted source**					
	Vaccinated	Willing	Undecided	Not willing	
TV	21.5	27.7	27.5	25.1	<0.0001
Radio	2.8	3.9	3.0	1.9	0.028
Health staff	78.4	68.2	59.1	52.8	<0.0001
Community health workers	11.9	11.5	5.5	5.9	<0.0001
Peers	6.9	6.7	6.8	8.2	0.176
Community leaders	4.9	5.7	4.9	3.7	0.050
Social media	23.3	17.7	20.2	22.1	0.003
SMS	4.3	1.9	3.2	3.0	0.001
Family	4.9	7.2	7.8	10.4	<0.0001
Work	9.1	2.8	4.4	3.6	<0.0001
Other	0.2	0.2	0.0	0.4	0.039
Internet	12.8	7.6	10.9	12.1	0.001
Private doctors/clinics	21.2	21.4	31.5	28.6	<0.0001
No one	0.0	0.0	0.1	0.7	<0.0001

**Table 5 vaccines-11-01109-t005:** Information needed to increase the uptake of the vaccines.

Info Needed to Increase the Uptake of Vaccines					Fisher’s Exact Test *p*-Value
	Vaccinated	Willing	Undecided	Not willing	
How vaccines work	18.8	20.3	15.3	14.9	<0.0001
Who is eligible?	8.7	9.1	8.7	8.0	0.768
What is inside the vaccines?	13.4	11.5	12.3	17.1	<0.0001
What is the difference between different types?	21.9	19.8	20.2	20.0	0.106
Side effects	43.3	45.5	44.6	42.0	0.362
The longevity of protection	22.8	19.0	25.7	20.8	0.003
Effectiveness	23.5	27.2	27.1	29.5	<0.0001
Effectiveness against new strains	23.8	16.3	19.9	18.2	<0.0001
Safety	30.3	35.9	34.8	42.4	<0.0001
The types of vaccines	10.8	8.1	5.9	8.1	<0.0001
Stats about vaccines	0.1	0.0	0.0	0.0	0.227
How they work on pregnant women	0.0	0.0	0.1	0.0	0.098
Other	0.1	0.0	0.1	0.0	0.377
Do not know	1.8	3.3	3.7	4.7	<0.0001
No answer	2.2	3.1	2.8	4.7	<0.0001
Nothing	0.1	0.0	0.0	0.1	0.329
Benefits of taking the vaccines					
	Vaccinated	Willing	Undecided	Not willing	
Feel protected	54.4	33.8	24.5	15.9	<0.0001
Family and friends feel protected	42.5	24.4	22.7	10.8	<0.0001
Motivated to do work	15.5	10.0	6.8	2.7	<0.0001
Motivated to socialize	11.8	8.7	6.8	2.3	<0.0001
Revert back to normal life	26.3	20.2	17.5	11.2	<0.0001
Travel	33.5	40.5	50.4	63.8	<0.0001
Less death	36.6	29.0	21.7	11.0	<0.0001
Saving money	19.1	16.1	12.6	10.4	<0.0001

## Data Availability

Data available upon request.

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
