# Peer review of "COVID-19 Vaccination Personas in Syria: Evidence from a Cross-Sectional Survey"

_vaccines, 2023, doi:10.3390/vaccines11061109_

Round 1

Reviewer 1 Report

The aim of the manuscript “COVID-19 Vaccination Personas in Syria: Evidence from a Cross-Sectional Survey” is to understand the main determinants of vaccination coverage in sample of 17,000 persons living in Syria.

The manuscript is well structured, the objectives are clearly stated, but the methods section should be improved as well as the presentation of results. Conclusions are consistent with the results.

Here comments and suggestions by section:

Methods

It is necessary to specify how the interviewed subjects have been recruited and the sampling methods adopted. Information about the survey methods (face-to-face, telephone, online) and instruments (structured questionnaire, semi-structured…) was lacking.

Results

Comments on results, such as “Given this professional composition of the sample, it is not, therefore, surprising that, on average, it is a very well-educated sample” should be moved to the discussion section.

The comparison of the selected characteristics has been done by using the chi-square test. However, the chi-square test cannot be used to test the differences between percentages, the authors should use absolute frequencies instead. In alternative , percentages can be compared using the Fisher’s exact test or the normal approximation to Fisher’s exact test (https://www.stat.berkeley.edu/~stark/SticiGui/Text/percentageTests.htm)

In table 4 the list of the source of information is duplicated with a different percentage distribution by vaccination status, it is not clear why.

Table 3 at page 6 should be renumerated (a table 3 already exists). The meaning of the scale (1 to 5) adopted  for question like “How challenging is to get the Covid-19 vaccine” is not clear (from easy = 1 to difficult = 5 ?)

The paragraph 3.3 seems to be dedicated to those “undecided” rather than “Not vaccinated and undecided”

Final consideration

The use of regression analysis may help to investigated the relationship between vaccine attitudes and socio-demographic characteristics.

In general, the manuscript provides interesting information about the profile of  vaccine hesitancy in a country with many humanitarian issues like Syria, but some major drawbacks should be fixed to consider the study for publication.

Author Response

Reviewer 1

Comments and Suggestions for Authors

The aim of the manuscript “COVID-19 Vaccination Personas in Syria: Evidence from a Cross-Sectional Survey” is to understand the main determinants of vaccination coverage in sample of 17,000 persons living in Syria. The manuscript is well structured, the objectives are clearly stated, but the methods section should be improved as well as the presentation of results. Conclusions are consistent with the results.

Response: We thank the reviewer for this positive feedback, and we believe that we have addressed all of the remaining comments as further suggested below.

Here comments and suggestions by section:

Methods

It is necessary to specify how the interviewed subjects have been recruited and the sampling methods adopted. Information about the survey methods (face-to-face, telephone, online) and instruments (structured questionnaire, semi-structured…) was lacking.

Response: We thank the reviewer for this comment. In the updated version of the manuscript, we have added further information on the survey methods, sampling as well as survey instruments. More specifically, we added the following paragraph to the paper: “Subjects eligible for receiving COVID-19 vaccine were randomly selected and enrolled in the study. Participants were mainly frontline workers from the health (health workers) and education sectors (teachers), randomly selected from healthcare centres and schools. In addition, key informants were selected from employees from related ministries. With that said, the survey relied on convenience sampling, meaning participants available at the time of the visit, and eligible to get the vaccine. The structured questionnaire used for data collection contained mainly multiple choice closed questions, with limited open-ended questions. Data was collected face-to-face upon visiting healthcare facilities, schools or government departments. Further break down of the sample size is provided in the results section.”

Results

Comments on results, such as “Given this professional composition of the sample, it is not, therefore, surprising that, on average, it is a very well-educated sample” should be moved to the discussion section.

Response: We have re-drafted the results section in order to provide only a description of the findings. Any allusion to possible discussion-related matters was removed from the Results section and placed under the discussion section.

The comparison of the selected characteristics has been done by using the chi-square test. However, the chi-square test cannot be used to test the differences between percentages, the authors should use absolute frequencies instead. In alternative , percentages can be compared using the Fisher’s exact test or the normal approximation to Fisher’s exact test (https://www.stat.berkeley.edu/~stark/SticiGui/Text/percentageTests.htm)

Response: We thank the reviewer for this comment. We have re-drafted this section of the methodology and it now reads: “The analysis used in this paper was descriptive in nature. The outcome variable was tabulated across the three sets of independent variables mentioned above in order to pro-vide an understanding of the link between various individual characteristics and willing-ness to vaccinate. The cross-tabulated percentages were compared using Fisher’s exact test and the p-values were reported”. In addition, we have re-worked the Results tables accordingly.

In table 4 the list of the source of information is duplicated with a different percentage distribution by vaccination status, it is not clear why.

Response:  We have provided further clarity to this. The upper part of the table refers to any source of information, while the bottom one to the most trusted source of information. This has been further clarified and rectified in the updated version of the manuscript.

Table 3 at page 6 should be renumerated (a table 3 already exists).

Response: We thank the reviewer for this comment. The tables have been re-ordered in the updated version of the manuscript. 

The meaning of the scale (1 to 5) adopted  for question like “How challenging is to get the Covid-19 vaccine” is not clear (from easy = 1 to difficult = 5 ?)

Response: We thank the reviewer for this comment. The information was collected in numeric terms, however, it broadly corresponds to the description above (1 – easy; 5 – difficult). We have provided further clarity on this when elaborating the results section of the paper.

The paragraph 3.3 seems to be dedicated to those “undecided” rather than “Not vaccinated and undecided”

Response: We thank the reviewer for this comment. For brevity, we have used only the term ‘undecided’. In the updated version of the manuscript, we used ‘not vaccinated and undecided’ throughout.

Final consideration

The use of regression analysis may help to investigated the relationship between vaccine attitudes and socio-demographic characteristics. In general, the manuscript provides interesting information about the profile of  vaccine hesitancy in a country with many humanitarian issues like Syria, but some major drawbacks should be fixed to consider the study for publication.

Response: We thank the reviewer for this comment. In the methods section of the paper, we have added the following paragraph: “Finally, as a robustness check, we have also conducted a preliminary logit modelling analysis. Four binary independent variables corresponding to the four vaccination personas were used in four separate logit modelling analyses. The four dependent variables were regressed against a battery of independent variables including socio-demographic variables as well as attitudes and beliefs regarding COVID-19 vaccines”.

Reviewer 2 Report

• The primary output/endpoint variable(s)/measurement(s) of the study should be defined.  • How was the sample size determined? This information should be explained in the Materials and Methods section. 

• Which sampling (probable or non-probable, etc.) method was used in the study? 

• Statistical tests for hypothesis testing and their assumptions should be specified in the study's statistical analysis in the Materials and Methods section. 

• The details (version, license number, etc.) of the statistical package(s) or program(s) should be given in the section of "Data Analysis or Statistical Analysis". 

• It should be explained how the qualitative and quantitative data are summarized under the sub-heading of Statistical Analyses in the Materials and Methods section of the study.  • Data analysis or Statistical analysis sub-section title should be added to the Materials and Methods.  • The exact P values should be added to the table(s) (e.g., p=0.25; p=0.03). 

• Which methods are used to model relationships between variables?

• The primary output/endpoint variable(s)/measurement(s) of the study should be defined.  • How was the sample size determined? This information should be explained in the Materials and Methods section. 

• Which sampling (probable or non-probable, etc.) method was used in the study? 

• Statistical tests for hypothesis testing and their assumptions should be specified in the study's statistical analysis in the Materials and Methods section. 

• The details (version, license number, etc.) of the statistical package(s) or program(s) should be given in the section of "Data Analysis or Statistical Analysis". 

• It should be explained how the qualitative and quantitative data are summarized under the sub-heading of Statistical Analyses in the Materials and Methods section of the study.  • Data analysis or Statistical analysis sub-section title should be added to the Materials and Methods.  • The exact P values should be added to the table(s) (e.g., p=0.25; p=0.03). 

• Which methods are used to model relationships between variables?

Author Response

Reviewer 2

Comments and Suggestions for Authors

  • The primary output/endpoint variable(s)/measurement(s) of the study should be defined. 

Response: We thank the reviewer for this comment. We have re-drafted the methods section, which now includes a further description of the main outcome variable used in the analysis. More specifically, we have added the following sentence to the manuscript: “This study is purely descriptive in nature, whereby we try to describe the main characteristics of the vaccination personas. In order to do that, our main outcome variable is a categorical variable, which captures the four vaccination personas: (a) those vaccinated; (b) those willing to be vaccinated; (c) those not willing to be vaccinated; and (d) those not sure if they want to be vaccinated. In order to distil the four personas, we relied on the following questions from the survey: “Have you received one or more Covid-19 vaccines?” and “How much would you like to receive the Covid-19 vaccine?”.

  • How was the sample size determined? This information should be explained in the Materials and Methods section. 

Response: We thank the reviewer for this comment. In the methods section of the paper, we have added the following paragraph, which, inter alia further elaborates on how the sample size was determined: “Subjects eligible for receiving COVID-19 vaccine were randomly selected and enrolled in the study. Participants were mainly frontline workers from the health (health workers) and education sectors (teachers), randomly selected from healthcare centres and schools. In addition, key informants were selected from employees from related ministries. With that said, the survey relied on convenience sampling, meaning participants available at the time of the visit, and eligible to get the vaccine. The structured questionnaire used for data collection contained mainly multiple choice closed questions, with limited open-ended questions. Data was collected face-to-face upon visiting healthcare facilities, schools or government departments. Further break down of the sample size is provided in the results section.”. It is also worth pointing out that logistical and budgetary considerations were involved when determining the final size of the sample.

  • Which sampling (probable or non-probable, etc.) method was used in the study? 

Response: We thank the reviewer for this comment. In the methods section of the paper, we have added a paragraph, which also further describes the sampling method adopted in the study: “Subjects eligible for receiving COVID-19 vaccine were randomly selected and enrolled in the study. Participants were selected by visiting healthcare centres. In addition, healthcare workers from selected healthcare centres were also selected. The survey participants also included teachers selected by visiting random schools. Finally, employees from visited ministries and related departments were also selected for the study. With that said, the survey relied on random sampling (selected from participants available at the time of the visit, and eligible to get the vaccine). The questionnaire used for data collection was a structured questionnaire, with limited open-ended questions in order to get as much feedback and precise answers from participants as possible. Data was collected face-to-face upon visiting healthcare facilities, schools or government departments. Further break down of the sample size is provided in the results section.”

  • Statistical tests for hypothesis testing and their assumptions should be specified in the study’s statistical analysis in the Materials and Methods section. 

Response: We thank the reviewer for this comment. The paper is exploratory and descriptive in nature. Against this background, in the methods section of the paper, we have added the following couple of paragraphs: “The analysis used in this paper was descriptive in nature. The outcome variable was tabulated across the three sets of independent variables mentioned above in order to pro-vide an understanding of the link between various individual characteristics and willing-ness to vaccinate. The cross-tabulated percentages were compared using Fisher’s exact test and the p-values were reported”.

  • The details (version, license number, etc.) of the statistical package(s) or program(s) should be given in the section of "Data Analysis or Statistical Analysis". 

Response: We thank the reviewer for this comment. In the methods section we have added the following sentence: “All analyses were conducted using Stata 17”.

  • It should be explained how the qualitative and quantitative data are summarized under the sub-heading of Statistical Analyses in the Materials and Methods section of the study. 

Response: We thank the reviewer for this comment. In the methods section of the paper, we have added the following couple of paragraphs: “The analysis used in this paper was descriptive in nature. The outcome variable was tabulated across the three sets of independent variables mentioned above in order to pro-vide an understanding of the link between various individual characteristics and willing-ness to vaccinate. The cross-tabulated percentages were compared using Fisher’s exact test and the p-values were reported.

Finally, as a robustness check, we have also conducted a preliminary logit modelling analysis. Four binary independent variables corresponding to the four vaccination personas were used in four separate logit modelling analyses. The four dependent variables were regressed against a battery of independent variables including socio-demographic variables as well as attitudes and beliefs regarding COVID-19 vaccines”.

  • Data analysis or Statistical analysis sub-section title should be added to the Materials and Methods. 

Response: We thank the reviewer for this comment. The sub-heading “Data analysis” was added to the section Materials and Methods.

  • The exact P values should be added to the table(s) (e.g., p=0.25; p=0.03). 

Response: We thank the reviewer for this comment. The exact p-values were included in the tables.

  • Which methods are used to model relationships between variables?

Response: As elaborated above, this is mainly a descriptive study using a cross-tabulation of selected variables. In addition, and as a robustness check, we have also resorted to using logit modelling regression. Against that background, we have added the following paragraph in the Methods section: “Finally, as a robustness check, we have also conducted a preliminary logit modelling analysis. Four binary independent variables corresponding to the four vaccination personas were used in four separate logit modelling analyses. The four dependent variables were regressed against a battery of independent variables including socio-demographic variables as well as attitudes and beliefs regarding COVID-19 vaccines”.

Reviewer 3 Report

I have the following comments

1. How the study participnts were selected?

2. How the survey was condcuted, i.e., online, telephonic etc

3. Study population seems to be highly biased as reflected in over representation of educated and vaccinated participants.  

4. Table 4 is placed before table 3

5. Table 4 is not clear. It comntains two halves- upper and lower . What si the difference between the two parts of this table.

English language need a lot of correction and improvement

Author Response

Reviewer 3

Comments and Suggestions for Authors

I have the following comments

Response: We thank the reviewer for these comments. We believe that we have adequately addressed them as further elaborated in the responses below.

  1. How the study participants were selected?

Response: We thank the reviewer for this comment. In the methods section we have added the following paragraph, which further answers the question above: “Subjects eligible for receiving COVID-19 vaccine were randomly selected and enrolled in the study. Participants were selected by visiting healthcare centres. In addition, healthcare workers from selected healthcare centres were also selected. The survey participants also included teachers selected by visiting random schools. Finally, employees from visited ministries and related departments were also selected for the study. With that said, the survey relied on random sampling (selected from participants available at the time of the visit, and eligible to get the vaccine). The questionnaire used for data collection was a structured questionnaire, with limited open-ended questions in order to get as much feedback and precise answers from participants as possible. Data was collected face-to-face upon visiting healthcare facilities, schools or government departments. Further break down of the sample size is provided in the results section”.

  1. How the survey was conducted, i.e., online, telephonic etc

Response: We thank the reviewer for this comment. In the methods section, we have added the following paragraph, which further explains the mode of data collection: “Subjects eligible for receiving COVID-19 vaccine were randomly selected and enrolled in the study. Participants were selected by visiting healthcare centres. In addition, healthcare workers from selected healthcare centres were also selected. The survey participants also included teachers selected by visiting random schools. Finally, employees from visited ministries and related departments were also selected for the study. With that said, the survey relied on random sampling (selected from participants available at the time of the visit, and eligible to get the vaccine). The questionnaire used for data collection was a structured questionnaire, with limited open-ended questions in order to get as much feedback and precise answers from participants as possible. Data was collected face-to-face upon visiting healthcare facilities, schools or government departments. Further break down of the sample size is provided in the results section.”

  1. Study population seems to be highly biased as reflected in over representation of educated and vaccinated participants.  

Response: We thank the reviewer for this comment. We are aware of this limitation and have further added the following sentence in the limitation paragraph (Discussion section of the paper): “Second, the 17,000 respondents, while randomly selected from visitors to healthcare facilities, teachers at school and employees of government agencies could not be counted as providing a representative set of adults in the country”.

  1. Table 4 is placed before table 3

Response: Tables have been reordered in the updated version of the manuscript.

  1. Table 4 is not clear. It contains two halves- upper and lower . What si the difference between the two parts of this table.

Response: We have provided further clarity to this. The upper part of the table refers to any source of information, while the bottom one to the most trusted source of information. This has been further clarified and rectified in the updated version of the manuscript.

Round 2

Reviewer 2 Report

Acceptable 

Acceptable 

Reviewer 3 Report

Dear Authors

Thanks for your modifications

Thnaks